# Effects of Thermal Gradients in High-Temperature Ultrasonic Non-Destructive Tests

**DOI:** 10.3390/s22072799

**Published:** 2022-04-06

**Authors:** Juliano Scholz Slongo, Jefferson Gund, Thiago Alberto Rigo Passarin, Daniel Rodrigues Pipa, Júlio Endress Ramos, Lucia Valeria Arruda, Flávio Neves Junior

**Affiliations:** 1Graduate School on Electrical Engineering and Applied Computer Science, Federal University of Technology—Paraná, Curitiba 80230-901, Brazil; jeffersongund@alunos.utfpr.edu.br (J.G.); passarin@utfpr.edu.br (T.A.R.P.); danielpipa@utfpr.edu.br (D.R.P.); lvrarruda@utfpr.edu.br (L.V.A.); neves@utfpr.edu.br (F.N.J.); 2Centro de Pesquisas, Desenvolvimento e Inovação Leopoldo Américo Miguez de Mello—CENPES/PETROBRAS, Rio de Janeiro 21941-915, Brazil; julio.ramos@petrobras.com.br

**Keywords:** ultrasonic NDT, phased array, high temperature, thermal gradients

## Abstract

Ultrasonic inspection techniques and non-destructive tests are widely applied in evaluating products and equipment in the oil, petrochemical, steel, naval, and energy industries. These methods are well established and efficient for inspection procedures at room temperature. However, errors can be observed in the positioning and sizing of the flaws when such techniques are used during inspection procedures under high working temperatures. In such situations, the temperature gradients generate acoustic anisotropy and consequently distortion of the ultrasonic beams. Failure to consider such distortions in ultrasonic signals can result, in extreme situations, in mistaken decision making by inspectors and professionals responsible for guaranteeing product quality or the integrity of the evaluated equipment. In this scenario, this work presents a mathematical tool capable of mitigating positioning errors through the correction of focal laws. For the development of the tool, ray tracing concepts are used, as well as a model of heat propagation in solids and an experimentally defined linear approximation of dependence between sound speed and temperature. Using the focal law correction tool, the relative firing delays of the active elements are calculated considering the temperature gradients along the sonic path, and the results demonstrate a reduction of more than 68% in the error of flaw positioning.

## 1. Introduction

Non-Destructive Tests (NDT) are techniques used in evaluating materials and equipment, without any alterations over the specimen, intending to identify faults or discontinuities that may negatively affect their use. When applied during manufacturing, construction, assembly, or maintenance stages, these techniques can increase the quality of products and services, reduce costs, and preserve the environment and life [1].

There are many techniques for NDT inspections that are more or less applicable according to the type of object or flaws to be inspected. For example, one can cite techniques based on eddy current, acoustic emission, radiography, gammagraphy, tightness, penetrating liquids, magnetic particles, ultrasound, thermography, and visual testing.

Considering all NDT techniques, those based on ultrasonic signals are among the most used. Their extensive application is justified by easy assemble of the equipment, penetration, and propagation characteristics of the ultrasonic signal, the capacity of acquiring, based on received ultrasound signals, information related to location, size, and orientation of flaws [2].

Although it is not difficult to configure and use an ultrasonic inspection system to obtain information about failures and discontinuities, the correct interpretation of the obtained signals and their quantitative analysis is challenging to ascertain [3]. Furthermore, the results obtained through ultrasonic inspection do not depend only on the performance of the applied ultrasonic system [4]. Instead, they depend on a series of factors related to the inspector (skill, experience, training, and qualification), to the object under evaluation (access, geometry, surface conditions, and sample temperature), to existing flaws in the object (type, shape, location, and position), and also on data analysis tools (signal processing, visualization in multiple modes, cursors, sizing tools, and reporting tools) available to carry out such inspection.

Periodic inspections and monitoring are required to detect flaws and guarantee the integrity of equipment and industrial plants in the oil, gas, and energy sectors. Moreover, it is profitable for companies that this equipment is inspected while in service, enabling routine checks and minimizing financial losses related to downtime and the logistics of shutting down and starting up the equipment or even the complete industrial plant [5].

Performing in-service ultrasonic NDT typically means inspecting materials or equipment under high-temperature conditions. In these situations, proper inspection techniques and specially developed equipment must be used, and signal processing methods that consider temperature gradients along the ultrasound path must be applied [5].

Considering the operational and technological challenges, there are two main problems linked to the application of ultrasonic systems in the inspection of specimens at high temperatures: damage to the ultrasonic inspection system and distortion of ultrasound signals [6,7,8,9].

Damage to the ultrasonic inspection system due to high-temperature exposure can occur directly in piezoelectric or other encapsulated elements [10]. Therefore, the direct application of a conventional ultrasonic probe in hot surfaces, that can cause the active elements depolarization and thermal expansion of the internal components, must be avoided [11]. Regarding probe damage, manufacturers of equipment for ultrasonic NDT report the failure of conventional phased array probes when reaching 80 °C. Therefore, this is the temperature threshold considered as being high temperature for ultrasonic NDT in this paper. For safe use, it is advised that probe temperature does not exceed 60 °C and, if it does, the exposure time must not be longer than a few seconds.

In this scenario, several papers have been presented aiming to develop transducers applicable in high-temperature ultrasonic NDT. To this end, the selection of appropriate piezoelectric elements has been made. In addition, specific production techniques and the definition of structural materials with favorable characteristics to the application at high temperatures have been studied [8,9,12]. Generally, difficulties related to thermal expansion of the active elements, high production cost, and especially difficulties in producing materials with stable piezoelectric and electrical properties under high-temperature conditions are reported.

Furthermore, thermal insulation mechanisms, such as waveguides and wedges, have been widely applied to protect the ultrasonic measurement equipment from heat damage. These structures, built with a material of low thermal conductivity, guide the ultrasound signals from the transducer to the object under inspection, and vice versa, in addition to promoting the necessary thermal insulation for the preservation of the equipment [13,14,15].

The development of new non-destructive inspection methods and signal processing techniques, able to cope with signal distortions, considering the dispersion and transmission characteristics of ultrasonic signals on the guides, wedges, or other installed apparatus, is as important as the thermal system protection [11,16,17]. However, in a state-of-the-art review, it is noticed that few papers have been presented in this context.

The temperature dependence of sound speed is a fundamental problem to be considered when performing ultrasonic NTD under high temperature conditions [7]. So, aiming to circumvent the signal distortions problems, the relationship between temperature and sound speed has been the object of study since the 1970s. [18,19]. Later, through investigations of ultrasound propagation inside hot materials, the sound speed decrease, as well as the attenuation of the signal amplitude, have been correlated with the specimen temperature increase [6,20].

The temperature gradients observed along the ultrasonic path modify the incidence angle of sound waves and cause variation of the sound propagating speed, besides propagation angle changes [7]. Thereby, the ultrasonic systems where the setup inspection, the focal laws in phased array systems, is defined considering homogeneous temperature along the wave path, will be subject to more uncertainties, measurement errors, and incorrect characterization of the flaws [7,21]. This fact is evident in the experimental tests presented by Huggett (2017) [22], in which the errors in terms of identifying the depth and diameter of side-drilled holes are increasingly more significant with the increase in the temperature of the body under evaluation. As a result, the correction of the focal laws is such a significant action when performing high-temperature inspections [21].

The first step to correct the focal laws is knowing the ultrasonic path inside the specimen in high temperature. In this direction, mathematical models, capable of predicting the trajectory of ultrasonic beams when they propagate in anisotropic media, are presented by Mineo, Lines, and Cerniglia (2021) [23] and Marvasti (2014) [21]. The model proposed by Mineo, Lines, and Cerniglia (2021) [23] is used to exemplify a welding process in which the region close to the weld joint is considered to have several layers at different temperatures. The results obtained demonstrate the deflection of ultrasonic beams as a function of temperature gradients in the metallic body. The model presented by Marvasti (2014) [21], in turn, is used to develop a method for correcting the distortions suffered by ultrasonic signals due to the presence of thermal gradients along the sonic path.

Although the effects of temperature gradients on NDT are reported in the bibliography, there is a significant gap to be filled to compensate for such distortions suffered by ultrasonic signals. Neither practical methods of focal laws correction or methods based on digital post-processing of signals are sufficiently developed.

In this direction, this paper presents a method to correct the focal laws in high temperature phased-array ultrasonic NDT. The implementation of the tool, which is based on ray tracing concepts, required a practical study to define the influence of temperature on the speed of sound propagation, as well as a mathematical model for transient heat propagation through an ultrasonic wedge, both presented in this paper. Finally, through tests in simulation environments, the effects of different inspection configurations are presented and discussed, in addition to the validation of concepts through the comparison between inspection reconstructed images with and without the use of the proposed method.

## 2. The Temperature Dependence of the Sound Speed and Its Effects in Ultrasonic NDT

The modeling of the sound speed behavior as a function of the propagation media temperature is the first step to derive a solution for ultrasound beam distortions due to temperature gradients [7]. The determination of media temperature distribution profiles inside the specimen, waveguides, or wedges, over the inspection time, is another necessary step.

Therefore, the next section brings out a study of the relationship between temperature and sound speed, while Section 2.2 presents the modeling and simulation of heat transfer in a polyetherimide wedge over time due to its contact with a metallic surface at high temperature. In addition, in order to evaluate the effects related to the presence of temperature gradients along the ultrasonic path, simulations of non-destructive tests are carried out and discussed in Section 2.3.

### 2.1. Study of Temperature and Sound Speed Relationship

The temperature dependence of sound speed is a fundamental feature to be studied to avoid the issues related to ultrasonic signals distortions by temperature gradients. Then, looking for a valid smooth spatial temperature variation, an approximately linear dependence is assumed for the temperature and acoustic speed relationship which can be represented by Equation (Equation 1) [21]. The parameter α means the sound speed variation rate due to the temperature *T* variation, while $V_{0}$ means the propagation speed in the temperature of T= 0 °C.
(1)$$V=V_{0}+\alpha \cdot T$$

Sound speed is unique for each material and depends on physical parameters such as density and elasticity, both temperature-dependent [24]. So, for each material tested, it is crucial to define the parameters $V_{0}$ and α in Equation (Equation 1).

To accurately estimate these parameters, it is necessary to minimize the temperature changes in the specimen while testing. Therefore, the conception of an experiment with uniform temperature on test bodies comes as a requisite. The experimental apparatus used in tests is presented in Figure 1 and consists of a thermostatic bath, able to perform external flow of fluid in controlled temperature, and a container, in which the thermal exchange between the specimen and the circulating liquid happens.

The experimental procedure consists of exposing the specimen to a controlled temperature liquid for a period long enough to provide a homogeneous temperature over all the in-test material. A period of 60 min, counted from the temperature stabilization instant, was adopted. Then, using the Ominiscan MX2 console, data are acquired and saved for offline processing. Next, the ultrasonic test setup is configured to measure the specimen thickness that, related to ultrasonic waves transit time, defines the mean acoustic speed through the material.

Two specific materials are tested: super duplex steel and polyetherimide. Super duplex steel was chosen because it is commonly used in the oil industry to build equipment that operates at high working temperatures, such as pressure vessels and multiphase separators. Polyetherimide was chosen because it is specially developed to manufacture high-temperature ultrasonic wedges due to its low thermal conductivity.

Both materials are assayed in temperatures between 10 °C and 80 °C, with successive increments of 10 °C. This temperature range, defined for the study of the relationship between temperature and sound speed, is wide enough for the results obtained to be used in defining the parameters of Equation (Equation 1). It stands out that the thermal expansion of the specimens is adequately measured and compensated.

The standard ASME20 block, presented in Figure 2a, is used for the super duplex steel tests, while the polyetherimide ones are performed using the SA32C-ULT-0L-IHC wedge from Olympus, shown in Figure 2b.

The analysis of the dataset is carried out with the Olympus NDT TomoView software. The necessary transit time of the ultrasonic signal to reach the opposite face of the specimen is acquired and combined with the specimen thickness, measured by a digital micrometer, resulting in sound speed values for the two materials under study. Table 1 presents the sound speed in super duplex steel and polyetherimide to each assayed temperature.

From the speed values computed at each assayed temperature, the parameters $V_{0}$ and α, given in Equation (Equation 1), are defined for both super duplex steel and polyetherimide by linear regression. The results are presented in Table 2.

As the relationship between the temperature and the sound speed for the materials is known, the next step is to draw the temperature profile along the ultrasonic signal path over the inspection time. So, in the next section, a transient heat transfer model in an ultrasonic inspection wedge is presented.

### 2.2. Heat Transfer in Ultrasonic Inspection Wedge

During a high-temperature ultrasonic inspection, heat transfer occurs, by conduction and convection, from the hot metal surface to the inspection system. In order to investigate the transient temperature distribution profile in the ultrasonic wedge during the inspection procedure, that is, from the moment the wedge is placed in contact with a heated metal surface, a heat transfer simulation model is designed in COMSOL software.

The SA32C-ULT-60L-IHC polyetherimide wedge is associated with a super duplex steel plate, and the entire setup is involved in an air volume for simulated modeling. The wedge and the steel plate are set at initials temperatures of 25 °C and 120 °C, respectively. The air volume is set to an initial temperature of 25 °C and moves at a speed of 0.05 m/s. The wedge temperature is defined as the ambient temperature, while the metallic surface temperature is defined with reference to the operating temperature of oil dehydrator tanks that are usually installed in oil exploration platforms. The model solution is based on the toolboxes *heat transfer in solids and fluids* and *laminar flow* in a three-dimensional mode. The other physical and thermal proprieties of super duplex steel and polyetherimide, necessary to the simulation model, are presented in Table 3.

Figure 3 shows the finite element mesh obtained with COMSOL software, and a temperature distribution frame during simulation. In Figure 3b we can see the air flux going from right to left in the image and the aspect of heat convection generated by the heated steel plate. In Figure 4, the temperature distribution in the outer faces of the wedge for different simulation times is shown, while Figure 5 presents the temperature gradients reached in a longitudinal section, taken in the center of the wedge, for the same simulation times. This central section is highlighted as it is taken as the plane through which the ultrasonic beams propagate.

From Figure 5, it is possible to observe that heat transfers predominantly by conduction, from the hot metallic body to the lower face of the wedge and towards its upper face. However, the heat transfer by convection is still observed, characterized by the heated fluid air close to the side faces of the wedge. This effect is more evident on the smallest side face of the wedge than on the bigger one due to the wind direction.

The temperature distribution in the wedge is simulated during time windows of 60 min. The generated data is used, combined with the results from the experimental tests presented in Section 2.1, as input to a phased-array-based ultrasonic NDT simulated study, presented in the next Section 2.3.

### 2.3. Simulation of NDT Procedures Subjects to Temperature Gradients

In order to evaluate the effects related to the presence of temperature gradients along the ultrasonic path, that is, to verify how speed changes and sonic beam distortion impact on NDT, if not adequately treated, simulations of non-destructive tests were carried out in the software CIVA.

As there is no possibility, at the present moment, to simulate at CIVA temperature gradients inside the wedge, the functionality *Variable Temperature or/and Speed* (VTS) in liquid medium was used in the wedge representation. This tool allows the simulation of mediums with changes in temperature, sonic speed, or both. In this paper, the variable temperature option is used.

The use of the VTS tool has, as a basic requirement, the definition of the specimen geometry. It is possible to choose several pre-established formats such as planar, cylindrical, conical, spherical, or a 2D or 3D model of the test body. In the present paper, a 2D model of an extended test body, composed by the wedge and the metal block, is used as a prototype. Figure 6a shows the prototype layout, its dimensions, and each segment. The chosen dimensions, as in the heat transfer model, are those of the Olympus SA32-ULT-N60L-IHC wedge. Once validated, the 2D model is extruded, resulting in wedge/metal block set, shown in Figure 6b.

The definition of an interface segment bisects the test body into two distinct pieces, to which it is possible to configure materials with different physical properties. In this paper, it is considered that the contact with the wedge will not significantly impact the temperature of the metallic test body, as this test body has much bigger dimensions compared with the wedge ones. So, the temperature of the steel block is set as homogeneous and constant.

The wedge, in turn, is chosen as a fluid with variable temperature. For this, it is necessary to specify the temperature distribution map throughout it. The temperature maps used in the NDT simulations are obtained through the heat propagation model presented in Section 2.2, and the maps are imported into CIVA after being formatted according to the pattern specified in the software user manual. Still, regarding the wedge representation in CIVA, it is necessary to choose the parameters that relate to the sound propagation speed and the temperature of the propagation medium. To this end, the parameters defined in the experimental tests presented in Section 2.1 are chosen.

The Olympus 5L64-A32, a 64-element Phased Array (PA) transducer, is selected to perform the simulations, as well as a linear sweep (electronic sweep) mode. The use of 16 active elements in each sequence was defined, with one element stepping between subsequent sequences, resulting in a total of 49 sequences. That is, elements 1 to 16 are used for the first sequence, elements 2 to 17 are used for the second sequence, and so on until sequence 49, for which elements 49 to 64 are used.

The refraction angle is set as $\theta_{r}=40{}^{\circ}$. The flaw to be analyzed is a side-drilled hole (SDH) centered in the coordinates x = 8 mm and z = 75 mm. The coordinates system origin is placed in the center of the PA transducer and the *z* coordinate is defined as perpendicular to the face of the probe. Figure 7 presents the final simulation setup, with the probe and flaw representation as well as the temperature distribution map throughout the wedge.

In Figure 7, the paths performed by ultrasonic beams emitted in three different positions, beginning, center, and end of the PA transducer, are highlighted. It is worthwhile to note that the incidence angles are different for all beams as each beam is subject to different temperature gradients. In the realized tests, it is also noteworthy that beams originating from the same position but at different times also have different incidence angles due to the variation of the sonic paths as a function of the temperature time variation.

After simulations, the data is processed, and several images are derived to evaluate the effects of different temperature gradients on image reconstruction. A total of four scenarios have been simulated. The first is used as a reference to establish location and positioning errors; it also represents the initial moment of inspection, corresponding to a situation for which there are no temperature gradients in the wedge, and this can be taken as a homogeneous body. The result of the reconstruction, in an E-Scan mode, can be seen in Figure 8. The cursors were positioned in the pixel of greater intensity and the flaw positioning indication, x=7.78 mm and z=74.63 mm, corresponds to the SDH edge.

Figure 9 shows the E-Scan images for the inspections simulated with the temperature gradients shown in Figure 5, which represents the temperature distribution profiles for 20, 40, and 60 min elapsed from start of inspection, respectively.

Note that the position of the flaw becomes more distant with the temperature growth. This effect is consistent with the decrease in sound propagation speed as a function of temperature increase, a fact that is described in the literature and verified experimentally in the tests presented in Section 2.1.

A second effect can be observed by comparing the sequence of images shown in Figure 9. It is the change in the slope of the flaw representation. This effect is associated with the deflection due to the ultrasound beam behavior as a function of the temperature anisotropy of the medium.

In order to verify the total flaw positioning error, the pixels of greater intensity have been identified in the reconstructed images for the simulations without temperature gradients and with gradients resulting from 60-minute exposure. Comparison of the results of Figure 8 and Figure 9c shows a vertical displacement of 2.55 mm and a horizontal displacement of 0.46 mm, resulting in a total displacement of 2.59 mm.

The results presented in this section show the need to consider temperature gradients along the sonic path to minimize errors resulting from distortions suffered by the ultrasonic beam in high temperatures NDT. Therefore, Section 3 presenteds a technique, based in ray tracing, to correct the focal laws and compensate the effects of thermal gradients throughout the ultrasonic wedge.

## 3. Ray Tracing Based Focal Laws Correction

A challenge in the development of models for ultrasonic phased array NDT concerns the propagation of the ultrasonic wave in heated media. The temperature gradients present in an inspected object cause variation in sound speed, which in turn cause deviations in the ultrasound trajectory along the material [5]. This phenomenon affects image reconstruction and may cause errors in the flaw positioning and characterization.

This can be mitigated if the image reconstruction algorithm takes into account trajectory deviations or if such distortions are compensated in the calculation of the focal laws. In both cases, it is necessary that the algorithms are previously supplied with information about the transit times or perform some kind of trajectory simulation themselves.

The estimation of transit time considering temperature gradients is essentially based on simulations, for which there are three predominant methods: finite elements, finite differences, and ray tracing [21].

Applying the concept of ray tracing, in media whose acoustic speed varies smoothly, the propagation of ultrasonic waves is described by a first-order ordinary differential equations (ODEs) system. By tracking a single point on a wavefront, the wavefront moves with a local speed V(x,z) in the direction determined by the angle ϕ with respect to the axis *x*. Thus, for each local sound speed, the longitudinal and perpendicular displacements is written as Equations (Equation 2) and (Equation 3) [21].
(2)$$\frac{dz}{dt}=V\left(x,z\right)\cdot sin\left(\phi \right)$$
(3)$$\frac{dx}{dt}=V\left(x,z\right)\cdot cos\left(\phi \right)$$
with a variation of direction given by:(4)$$\frac{d\phi }{dt}=\hat{i}\cdot \vec{\nabla }V\left(x,z\right)\cdot sin\left(\phi \right)-\hat{k}\cdot \vec{\nabla }V\left(x,z\right)\cdot cos\left(\phi \right)$$
where $\vec{\nabla }$ is the gradient operator, and $\hat{i}$ and $\hat{k}$ are the unit vectors in the direction of the axis *x* and *z*, respectively.

By solving the Equations (Equation 2)–(Equation 4), it is possible to determine not only the path taken by the ultrasonic wavefront, but also the transit time values needed to specify the relative firing delays for each PA active element, enabling the definition of focal laws so that the effects of temperature gradients are mitigated.

In this work, the ODEs system is solved with the software MATLAB and the results are used to calculate focal laws for situations where inspection is made at high temperatures. In such cases, the propagation media are subject to thermal gradients and, consequently, propagation speed variations and distortions of the ultrasonic beans are observed and must be corrected.

The Figure 10 presents two sequences of the focal law calculated through the proposed focal law correction tool, for a refracted angle of 40, compared to the same sequences of the focal law obtained by conventional tool, which disregard the presence of temperature gradients. The relative firing time delays of the first sequence (elements 1 to 16) and the last sequence (elements 49 to 64) of the linear scan are illustrated. The curve in black represent the relative firing time delays for conventional focal law calculator tool, and the curve in red represents the relative firing time delays obtained through the tool that considers the temperature gradients along the sonic path when performing the calculation of the focal law.

Notably, in Figure 10 the delay times are expressed in relation to the last of the 16 elements used to form the plane wave. This is because it takes longer for the wave emitted by the last element of the aperture to travel from the transducer to the plane wavefront selected for calculation. Therefore, this element must be fired first, at a time defined as t=0 s. Thus, all other elements are then triggered at their respective relative delay times with respect to this reference element.

Other refraction angles were assayed and, for all of them, it is observed that the use of the last 16 elements of the phased array transducer implies the need for a minor correction of the Focal Laws. It is also noted that the choice of smaller refraction angles also implies a smaller need for correction, as summarized in Table 4.

From the application of the proposed method, the focal laws could be imported into the CIVA and, in order to evaluate the ability to correct positioning errors related to the presence of temperature gradients along the sonic trajectory, new simulations could be carried out.

Figure 11 presents a comparison between the reconstructed E-scan inspection image for the case where temperature gradients along the sonic path are not considered (Figure 11a) and the reconstructed inspection image after proper treatment of the thermal gradients through application of the corrections proposed in this work (Figure 11b).

The positioning errors observed after applying the focal law correction method in the *x* and *z* directions are 0.33 mm and 0.75 mm, respectively, resulting in a total positioning error of 0.82 mm. In this way, the application of the proposed focal law correction method, implemented and tested through simulation, brings an improvement in the positioning of the discontinuity, resulting in an approximate decrease of 68% of the error associated with temperature gradients along the sonic path.

When looking at Figure 11b, it is possible to notice a certain distortion in the failure representation. This is due to the fact that the correction is performed beam by beam and that the correction capacity is directly impacted by the origin of each sound wave and by the gradients to which it is submitted. Thus, the correction for certain beams does not present the same efficiency as for the others, resulting in distortions in the reconstructed image. To minimize this effect, the calculation routines are being refined in order to define the optimal parameters in terms of the integration step.

## 4. Conclusions

The deflection caused in ultrasonic beams due to temperature gradients along the sonic path is one of the problems associated with applying ultrasonic techniques for non-destructive tests at high temperatures. In order to mitigate this effect, it is necessary to know the temperature distribution in the ultrasonic wedge during the inspection procedure.

Therefore, a model of heat propagation throughout the polyetherimide wedge was developed. In order to contemplate the heat transfer by conduction and convection, the *heat transfer in solids and fluids* and *laminar flow* COMSOL toolboxes were applied. The heat propagation was simulated for a total of 60 min, and the resulting heat distributions were used to simulate NDT subject to temperature gradients.

Knowing the temperature distributions throughout the wedge, it is also necessary to define the relationship between the propagation speed of the ultrasonic wave and the temperature of the medium in which it propagates. This relationship was experimentally evaluated for the polyetherimide wedge and super duplex steel. The sound speed, in both media, was tested for a range between 10 °C and 80 °C. The results obtained were used in a linear regression process to define the linear relationship between propagation speed and temperature for the materials tested. The experimental results showed a sound–propagation speed decrease due to an increase in medium temperature at rates consistent with those presented in the literature.

The relationship between propagation speed and wedge temperature and the modeled temperature distribution were then used to simulate NDT subjects to temperature gradients in the CIVA. The obtained simulation results have shown an increase in the positioning error with the increase in the wedge exposure to heat from the metallic body under inspection. After 60 min of inspection, a positioning error of 2.59 mm was observed comparing the results obtained with and without temperature gradients. Furthermore, due to ultrasonic beam deflection, a change in the slope of the flaw representation was also observed.

Using the focal law correction tool, which is based on ray tracing concepts, the relative excitation delays of the active elements were calculated considering the temperature gradients along the sonic path and, in this way, mitigating the distortion effects of the sonic beam as a function of the thermal anisotropy of the material.

The corrected focal laws were then formatted and imported into the CIVA software so that, through simulated tests, the effectiveness of the technique could be verified. The results demonstrate a reduction of more than 68% in the error of flaw positioning compared to the situation in which the temperature gradients are disregarded. The method, however, by promoting beam-to-beam correction, causes some distortion in the reconstructed image.

In general, the results presented in this work show the need to consider temperature gradients along the sonic path to minimize errors resulting from distortions suffered by the ultrasonic beam in high-temperature NDT.

The methodology presented in this paper can be used to solve similar cases by using the following steps:Implement the heat propagation model in dedicated software, defining the wedge’s geometry as well as its physical and thermal properties. Boundary conditions and solution parameters of the thermal model must also be defined. At the end of the simulation, it is recommended to use the longitudinal section taken at the center of the wedge as the plane through which the ultrasonic beams propagate;Experimentally define the relationship between the speed of sound propagation and the temperature of the medium under study. To do this, ensure homogeneous temperature along the test body and measure the propagation velocity with an appropriate ultrasonic device. Consider the thermal expansion of the test body in order to obtain more accurate results;Apply ray tracing concepts for the definition of the sonic paths and the transit times necessary for the generation of a plane wave and, thus, proceed the definition of the appropriate focal laws. The implementation might be easier by starting the solution from points sampled on a straight line inside the test body, with slope defined by the desired angle of refraction;Simulate cases of phased array ultrasonic inspection at homogeneous temperature and at high temperatures (using the modeled temperature profiles) in a tool that allows the import of the corrected focal laws.

Future works on high-temperature ultrasonic inspection techniques point to the validation, in practical tests, using certified specimens, of the results presented in this paper and, also, to develop a complementary procedure aiming the minimization of distortions that can appear in the reconstructed images.

## Figures and Tables

**Figure 1 sensors-22-02799-f001:**
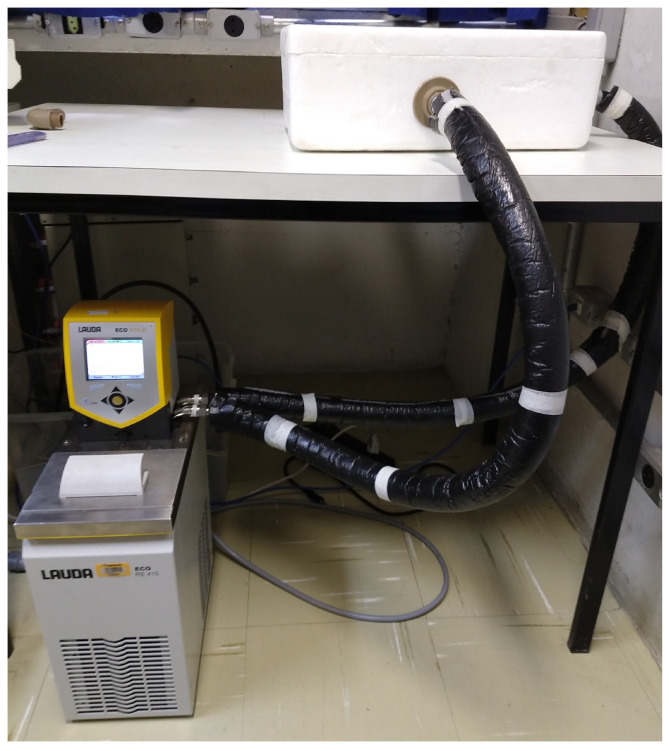
Experimental apparatus used in the tests to determine the temperature dependence of the sound speed. The setup consists of a thermostatic bath, capable of promoting the external flow of temperature-controlled fluid, and a container where the thermal exchange happens.

**Figure 2 sensors-22-02799-f002:**
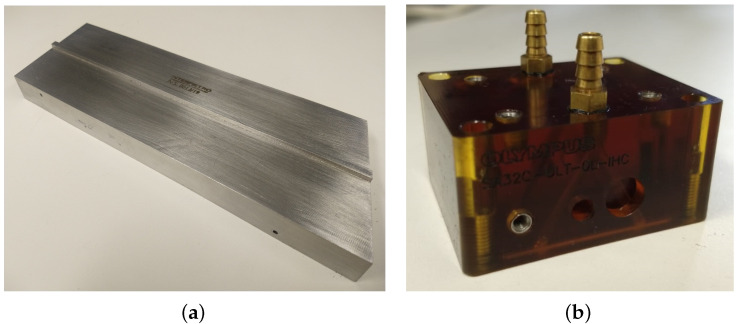
Specimens used in the tests to determine the thermal dependence of the sound speed: (**a**) Super duplex steel block and (**b**) Olympus SA32C-ULT-0L-IHC wedge in polyetherimide.

**Figure 3 sensors-22-02799-f003:**
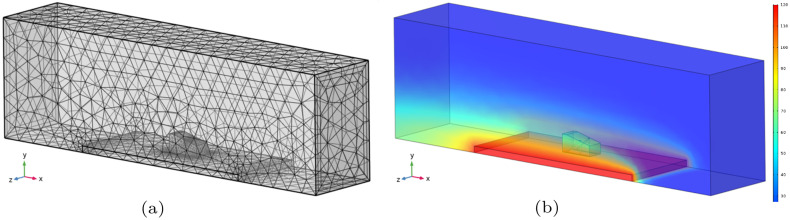
Heat transfer model in COMSOL: (**a**) Finite element mesh created for model solution; (**b**) Frame of the temperature distribution taken in the simulation time of 2.5 min.

**Figure 4 sensors-22-02799-f004:**
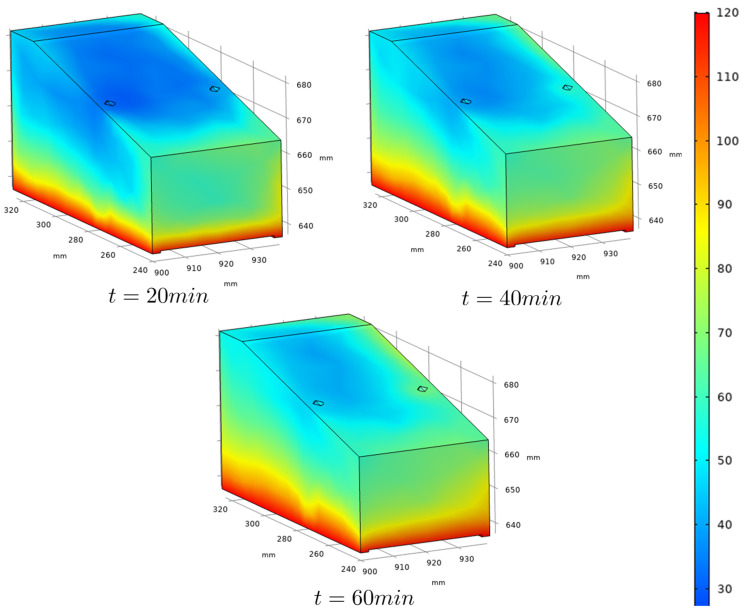
Simulated frames showing the temperature distribution in the outer faces of the wedge for three specific times. The heat propagation to the upper regions of the wedge can be observed.

**Figure 5 sensors-22-02799-f005:**
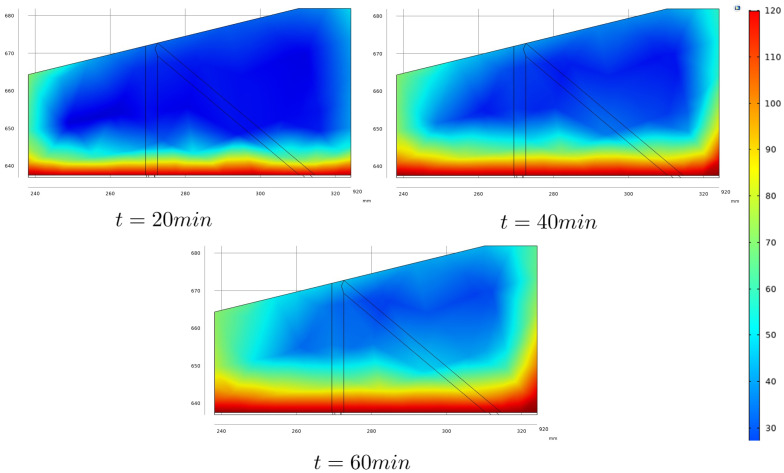
Simulated frames showing the temperature distribution in a longitudinal section taken in the center of the wedge for three specific times. They will be used as temperature distribution maps in the high-temperature ultrasonic NDT simulation presented in Section 2.3.

**Figure 6 sensors-22-02799-f006:**
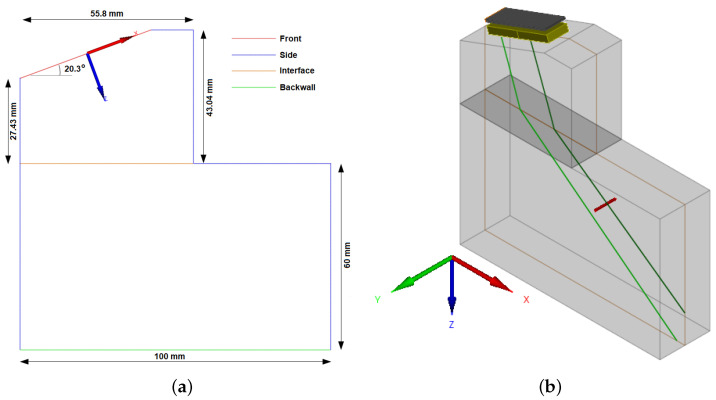
Extended test body, formed by the wedge and the metal block, configured in software CIVA to make it possible to simulate ultrasonic NDT subjects to temperature gradients. (**a**) 2D CAD and (**b**) test body resulting from the extrusion of the 2D model.

**Figure 7 sensors-22-02799-f007:**
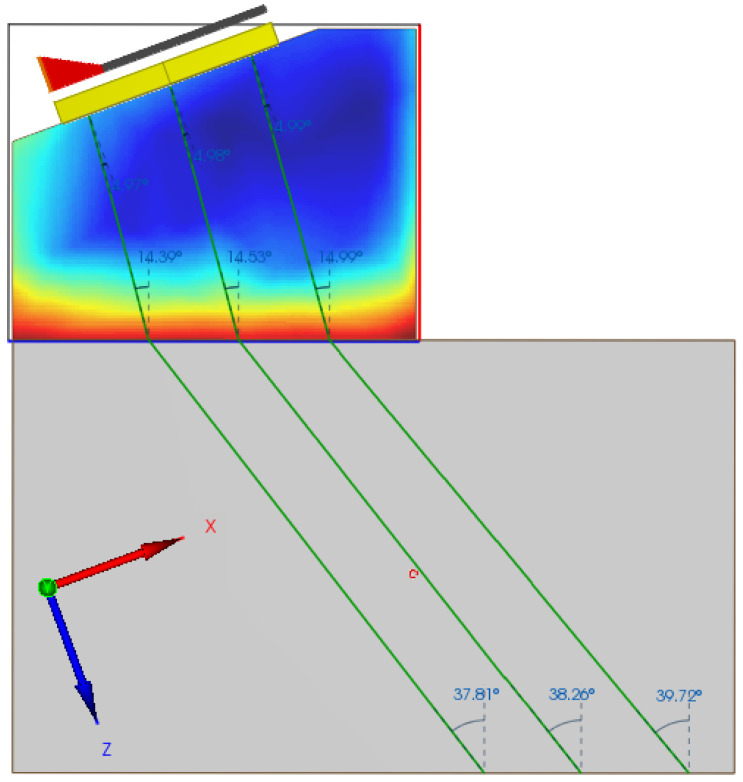
A side view of the extended test body, with temperature gradients throughout the wedge and the definition of an SDH-type flaw. Emphasis on the beams incidence angles for the ultrasonic emission from different positions in the PA transducer.

**Figure 8 sensors-22-02799-f008:**
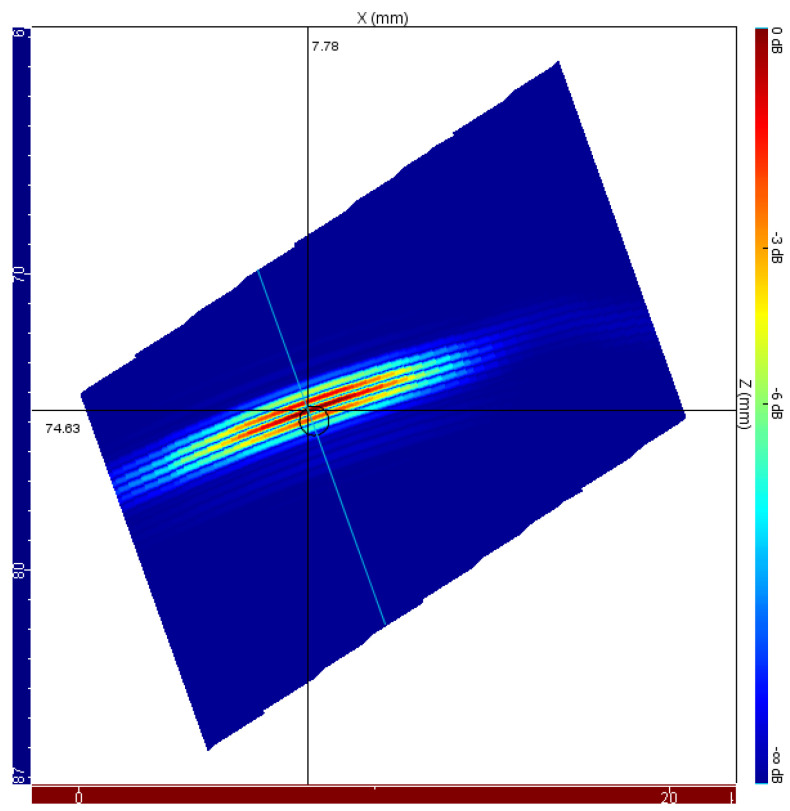
Ultrasonic E-Scan image of the side-drilled hole for the case with no temperature gradients throughout the wedge. It is taken in the initial instant of the inspection procedure and is used as a reference to establish location and positioning errors due to high-temperature effects.

**Figure 9 sensors-22-02799-f009:**
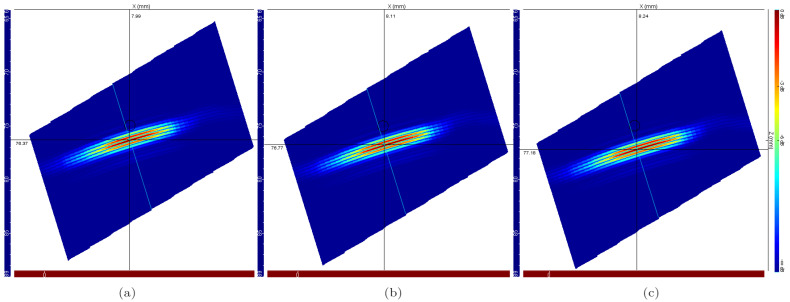
Ultrasonic E-Scan images of the side-drilled hole for the case with temperature gradients throughout the wedge. The results were obtained for the exposure of the ultrasonic system to high temperatures for (**a**) 20 min, (**b**) 40 min, and (**c**) 60 min.

**Figure 10 sensors-22-02799-f010:**
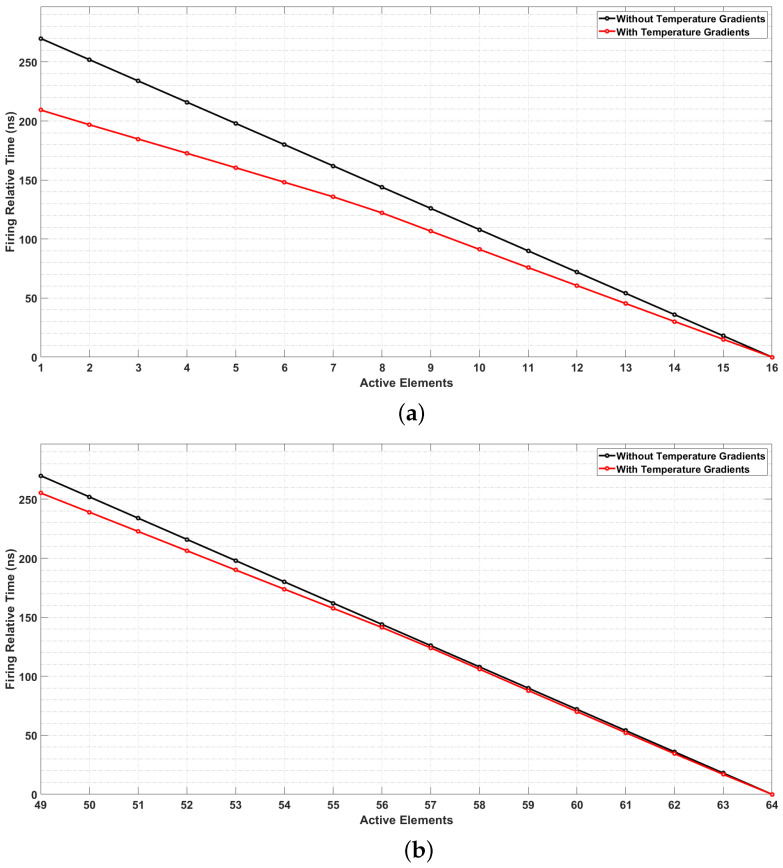
Focal Laws defined for generating a plane wave with an angle of refraction 40 in inspection procedures with and without temperature gradients along the sonic path. Wavefront formed (**a**) using the first 16 active elements of the PA transducer and (**b**) using the last 16 active elements of the PA transducer.

**Figure 11 sensors-22-02799-f011:**
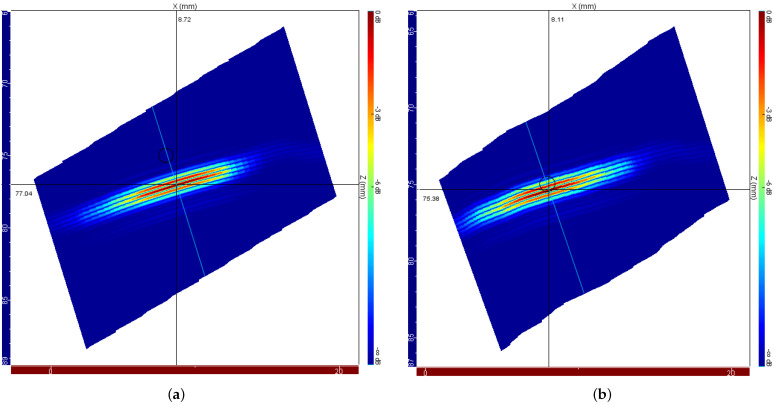
Ultrasonic E-Scan images of the side-drilled hole for the case where the temperature gradients along the sonic path are (**a**) disregarded and (**b**) appropriately treated by applying the proposed focal law correction methodology.

**Table 1 sensors-22-02799-t001:** Super duplex steel and polyetherimide sound speeds under different temperatures.

Temperature	Super Duplex Stell Sound Speed (m/s)			Polyetherimide Sound Speed (m/s)		
(°C)	Test 1	Test 2	Test 3	Test 1	Test 2	Test 3
10	6018.7	6009.7	6036.6	2470.4	2470.4	2469.1
20	5972.7	5965.6	6024.0	2448.6	2447.4	2447.4
30	5935.9	5935.8	5936.0	2433.1	2431.9	2431.9
40	5923.1	5923.2	5923.1	2417.8	2419.0	2419.0
50	5880.4	5884.6	5923.7	2401.5	2403.8	2405.0
60	5846.9	5872.7	5920.4	2391.2	2393.5	2394.6
70	5852.2	5864.9	5805.5	2383.2	2384.4	2387.8
80	5852.5	5848.3	5865.2	2354.0	2357.4	2355.3

**Table 2 sensors-22-02799-t002:** Parameters $V_{0}$ and α defined, to super duplex steel and polyetherimide, by linear regression.

Material	$V_{0}$ (m/s)	α (m/s/°C)
Super Duplex Steel	6030	−2.491
Polyetherimide	2480	−1.484

**Table 3 sensors-22-02799-t003:** Physical and thermal proprieties of super duplex steel and polyetherimide used in the transient heat transfer model in COMSOL.

Material	Density (kg/m^3^)	Thermal Conductivity (W/m/K)	Thermal Capacity (J/kg/K)
Super Duplex Stell	7820	17	450
Polyetherimide	1270	0.036	1890

**Table 4 sensors-22-02799-t004:** Comparison of the maximum correction times of the Focal Laws when using the first 16 (1–16) and the last 16 (49–64) active elements of the phased array transducer for different refraction angles $\theta_{r}$.

Refraction Angle	Active Elements	Maximum Correction Time (ns)
60°	1–16	58.8
	49–64	13.3
50°	1–16	58.0
	49–64	12.7
40°	1–16	56.5
	49–64	10.5
30°	1–16	54.7
	49–64	6.1

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
