# Peer review of "Effects of Thermal Gradients in High-Temperature Ultrasonic Non-Destructive Tests"

_sensors, 2022, doi:10.3390/s22072799_

Round 1

Reviewer 1 Report

The problem of taking into account the temperature distribution in the prism is an urgent task for the practice of ultrasonic nondestructive testing.

In the section "3. Ray Tracing Based Focal Laws Correction" in the formulas sen should be replaced with sin. In the reference [5] there are no formulas (2) - (4). It is necessary to specify the correct reference from where formulas (2) - (4) are taken.

68.34% - excessive precision!

The article would be logically complete if the results of application of the proposed approach in model experiments were shown. After all, these experiments can be carried out on the installation, which was used to study the effect of temperature on the velocity of longitudinal waves in the prism and control object. I do not insist on finalizing the article (I liked it!), but if it were done, the article would be very interesting and valuable.

Author Response

The authors appreciate your comments. The answers for each point are in the attachment. Please see it.

Reviewer 2 Report

The manuscript (sensors-1642879) studies the high-temperature thermal gradients based on ultrasonic. The authors gave the relationship of sound speed and medium at essayed temperatures and proposed that using ray tracing to define the reflection angle. It is well written and organized, so I agree to accept it after the following issues addressed.

1 In line34, is it qualitative analysis or quantitive analysis?
2 In line61, “greater” should be changed into “longer”.
3 In line 64, “characterization” should be deleted.
4 In table 1, do you consider the effect of material expansion, which will change the distance of the sonic propagation? Then, the speed of sonic will be influenced.
5. It is confused about the temperature in this manuscript. Authors measurement is between 10 – 80℃, but in line 186 you declared the temperature range is 25 – 120 ℃.
6. Could you explain “the 49 sequence” in line 241?
7. In line291, “transit times” should be corrected to “transit time”.
8. Could the authors add some description about Fig.10. It is obscure.
9. I don’t agree the authors’ opinion about the high-temperature. Its max temperature region for simulation and measurement only approaches 100 ℃. But it won’t be convinced by the audiences.

Author Response

(The authors gave the same response as above.)
